# QUADTREE ATTENTION FOR VISION TRANSFORMERS

**Shitao Tang**[1*]**, Jiahui Zhang**[2*]**Siyu Zhu**[2]**, Ping Tan**[12]
[1]Simon Fraser University, [2]Alibaba A.I. Lab
`shitaot@sfu.ca, zjhthu@gmail.com,`
`siting.zsy@alibaba-inc.com, pingtan@sfu.ca`

### ABSTRACT

Transformers have been successful in many vision tasks, thanks to their capability of capturing long-range dependency. However, their quadratic computational complexity poses a major obstacle for applying them to vision tasks requiring dense predictions, such as object detection, feature matching, stereo, etc. We introduce QuadTree Attention, which reduces the computational complexity from quadratic to linear. Our quadtree transformer builds token pyramids and computes attention in a coarse-to-fine manner. At each level, the top $K$ patches with the highest attention scores are selected, such that at the next level, attention is only evaluated within the relevant regions corresponding to these top $K$ patches. We demonstrate that quadtree attention achieves state-of-the-art performance in various vision tasks, e.g. with 4.0% improvement in feature matching on ScanNet, about 50% flops reduction in stereo matching, 0.4-1.5% improvement in top-1 accuracy on ImageNet classification, 1.2-1.8% improvement on COCO object detection, and 0.7-2.4% improvement on semantic segmentation over previous state-of-the-art transformers. The codes are available at https://github.com/Tangshitao/QuadtreeAttention.

## 1 INTROCUTION

Transformers can capture long-range dependencies by the attention module and have demonstrated tremendous success in natural language processing tasks. In recent years, transformers have also been adapted to computer vision tasks for image classification (Dosovitskiy et al., 2020), object detection (Wang et al., 2021c), semantic segmentation (Liu et al., 2021), feature matching (Sarlin et al., 2020), and stereo (Li et al., 2021), etc. Typically, images are divided into patches and these patches are flattened and fed to a transformer as word tokens to evaluate attention scores. However, transformers have quadratic computational complexity in terms of the number of tokens, i.e. number of image patches. Thus, applying transformers to computer vision applications requires careful simplification of the involved computation.

To utilize the standard transformer in vision tasks, many works opt to apply it on low resolution or sparse tokens. ViT (Dosovitskiy et al., 2020) uses coarse image patches of $16 \times 16$ pixels to limit the number of tokens. DPT (Ranftl et al., 2021) up-samples low-resolution results from ViT to high resolution maps to achieve dense predictions. SuperGlue (Sarlin et al., 2020) applies transformer on sparse image keypoints. Focusing on correspondence and stereo matching applications, Germain et al. (2021) and Li et al. (2021) also apply transformers at a low resolution feature map.

However, as demonstrated in several works (Wang et al., 2021c; Liu et al., 2021; Sun et al., 2021; Li et al., 2021; Shao et al., 2020), applying transformers on high resolution is beneficial for a variety of tasks. Thus, many efforts have been made to design efficient transformers to reduce computational complexity. Linear approximate transformers (Katharopoulos et al., 2020; Wang et al., 2020) approximate standard attention computation with linear methods. However, empirical studies (Germain et al., 2021; Chen et al., 2021) show those linear transformers are inferior in vision tasks. To reduce the computational cost, the PVT (Wang et al., 2021c) uses downsampled keys and values, which is harmful to capture pixel-level details. In comparison, the Swin Transformer (Liu et al.,

---

*Equal contribution

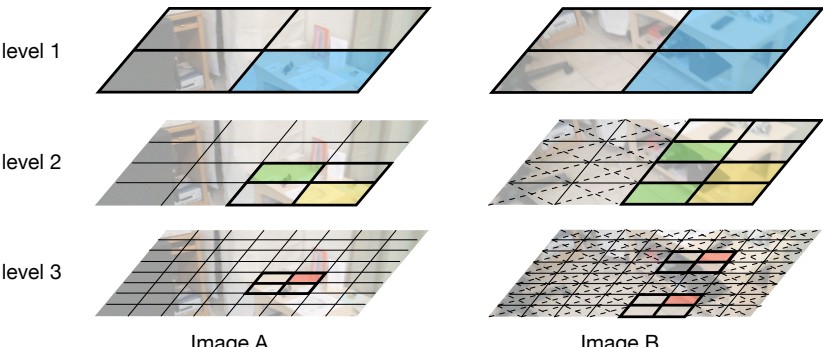

Figure 1: Illustration of QuadTree Attention. Quadtree attention first builds token pyramids by down-sampling the query, key and value. From coarse to fine, quadtree attention selects top $K$ (here, $K = 2$) results with the highest attention scores at the coarse level. At the fine level, attention is only evaluated at regions corresponding to the top $K$ patches at the previous level. The query sub-patches in fine levels share the same top $K$ key tokens and coarse level messages, e.g., green and yellow sub-patches at level 2 share the same messages from level 1. We only show one patch in level 3 for simplicity.

2021) restricts the attention in local windows in a single attention block, which might hurt long-range dependencies, the most important merit of transformers.

Unlike all these previous works, we design an efficient vision transformer that captures both fine image details and long-range dependencies. Inspired by the observation that most image regions are irrelevant, we build token pyramids and compute attention in a coarse to fine manner. In this way, we can quickly skip irrelevant regions in the fine level if their corresponding coarse level regions are not promising. For example, as in Figure 1, at the 1st level, we compute the attention of the blue image patch in image A with all the patches in image B and choose the top $K$ (here, $K = 2$) patches which are also highlighted in blue. In the 2nd level, for the four framed sub-patches in image A (which are children patches of the blue patch at the 1st level), we only compute their attentions with the sub-patches corresponding to the top $K$ patches in image B at the 1st level. All the other shaded sub-patches are skipped to reduce computation. We highlight two sub-patches in image A in yellow and green. Their corresponding top $K$ patches in image B are also highlighted in the same color. This process is iterated in the 3rd level, where we only show the sub-sub-patches corresponding to the green sub-patch at the 2nd level. In this manner, our method can both obtain fine scale attention and retain long-range connections. Most importantly, only sparse attention is evaluated in the whole process. Thus, our method has low memory and computational costs. Since a quadtree structure is formed in this process, we refer to our method as QuadTree Attention, or QuadTree Transformer.

In experiments, we demonstrate the effectiveness of our quadtree transformer in both tasks requiring cross attention, e.g. feature matching and stereo, and tasks only utilizing self-attention, e.g. image classification and object detection. Our method achieves state-of-the-art performance with significantly reduced computation, comparing to relevant efficient transformers (Katharopoulos et al. (2020); Wang et al. (2021c); Liu et al. (2021)). In feature matching, we achieve 61.6 AUC@20° in ScanNet (Dai et al., 2017), 4.0 higher than the linear transformer (Katharopoulos et al., 2020) but with similar flops. In stereo matching, we achieve a similar end-point-error as standard transformer, (Li et al., 2021) but with about 50% flops reduction and 40% memory reduction. In image classification, we achieve 84.0% top-1 accuracy in ImageNet (Deng et al., 2009), 5.7% higher than ResNet152 (He et al., 2016) and 1.0% higher than the Swin Transformer-S (Liu et al., 2021). In object detection, our QuadTree Attention + RetinaNet achieves 47.9 AP in COCO (Lin et al., 2014), 1.8 higher than the backbone PVTv2 (Wang et al., 2021b) with fewer flops. In semantic segementation, QuadTree Attention improves the performance by 0.7-2.4%.

## 2 RELATED WORK

**Efficient Transformers.** Transformers have shown great success in both natural language processing and computer vision. Due to the quadratic computational complexity, the computation of

full attention is unaffordable when dealing with long sequence tokens. Therefore, many works design efficient transformers, aiming to reduce computational complexity (Katharopoulos et al., 2020; Choromanski et al., 2020; Shao et al., 2021; Wang et al., 2020; Lee et al., 2019; Ying et al., 2018). Current efficient transformers can be categorized into three classes. 1) Linear approximate attention (Katharopoulos et al., 2020; Choromanski et al., 2020; Wang et al., 2020; Beltagy et al., 2020; Zaheer et al., 2020) approximates the full attention matrix by linearizing the softmax attention and thus can accelerate the computation by first computing the product of keys and values. 2) Inducing point-based linear transformers (Lee et al., 2019; Ying et al., 2018) use learned inducing points with fixed size to compute attention with input tokens, thus can reduce the computation to linear complexity. However, these linear transformers are shown to have inferior results than standard transformers in different works (Germain et al., 2021; Chen et al., 2021). 3) Sparse attention, including Longformer (Beltagy et al., 2020), Big Bird (Zaheer et al., 2020), etc, attends each query token to part of key and value tokens instead of the entire sequence. Unlike these works, our quadtree attention can quickly skip the irrelevant tokens according to the attention scores at coarse levels. Thus, it achieves less information loss while keeps high efficiency.

**Vision Transformers.** Transformers have shown extraordinary performance in many vision tasks. ViT (Dosovitskiy et al., 2020) applies transformers to image recognition, demonstrating the superiority of transformers for image classification at a large scale. However, due to the computational complexity of full attention, it is hard to apply transformers in dense prediction tasks, e.g. object detection, semantic segmentation, etc. To address this problem, Swin Transformer (Liu et al., 2021) restricts attention computation in a local window. Focal transformer (Yang et al., 2021) uses two-level windows to increase the ability to capture long-range connection for local attention methods. Pyramid vision transformer (PVT) (Wang et al., 2021c) reduce the computation of global attention methods by downsampling key and value tokens. Although these methods have shown improvements in various tasks, they have drawbacks either in capturing long-range dependencies (Liu et al., 2021) or fine level attention (Wang et al., 2021c). Different from these methods, our method simultaneously capture both local and global attention by computing attention from full image levels to the finest token levels with token pyramids in one single block. Besides, the K-NN transformers (Wang et al., 2021a; Zhao et al., 2019) aggregate messages from top $K$ most similar tokens as ours, but they compute the attention scores among all pairs of query and key tokens, and thus still has quadratic complexity.

Beyond self-attention, many tasks can largely benefit from cross attention. Superglue (Sarlin et al., 2020) processes detected local descriptors with self- and cross attention and shows significant improvement in feature matching. Standard transformers can be applied in SuperGlue because only sparse keypoints are considered. SGMNet (Chen et al., 2021) further reduces the computation by attending to seeded matches. LoFTR (Sun et al., 2021) utilizes linear transformer (Katharopoulos et al., 2020) on low-resolution feature maps to generate dense matches. For stereo matching, STTR (Li et al., 2021) applies self- and cross attention along epipolar lines and reduces the memory by gradient checkpointing engineering techniques. However, due to the requirement of processing a large number of points, these works either use linear transformers, which compromise performance, or a standard transformer, which compromises efficiency. In contrast, our transformer with quadtree attention achieves a significant performance boost compared with linear transformer or efficiency improvement compared with standard transformer. Besides, it can be applied to both self-attention and cross attention.

## 3 METHOD

We first briefly review the attention mechanism in transformers in Section 3.1 and then formulate our quadtree attention in Section 3.2.

### 3.1 ATTENTION IN TRANSFORMER

Vision transformers have shown great success in many tasks. At the heart of a transformer is the attention module, which can capture long-range information between feature embeddings. Given two image embeddings $\mathbf{X}_1$ and $\mathbf{X}_2$, the attention module passes information between them. Self-attention is the case when $\mathbf{X}_1$ and $\mathbf{X}_2$ are the same, while cross attention covers a more general situation when $\mathbf{X}_1$ and $\mathbf{X}_2$ are different. It first generates the query $\mathbf{Q}$, key $\mathbf{K}$, and value $\mathbf{V}$ by the

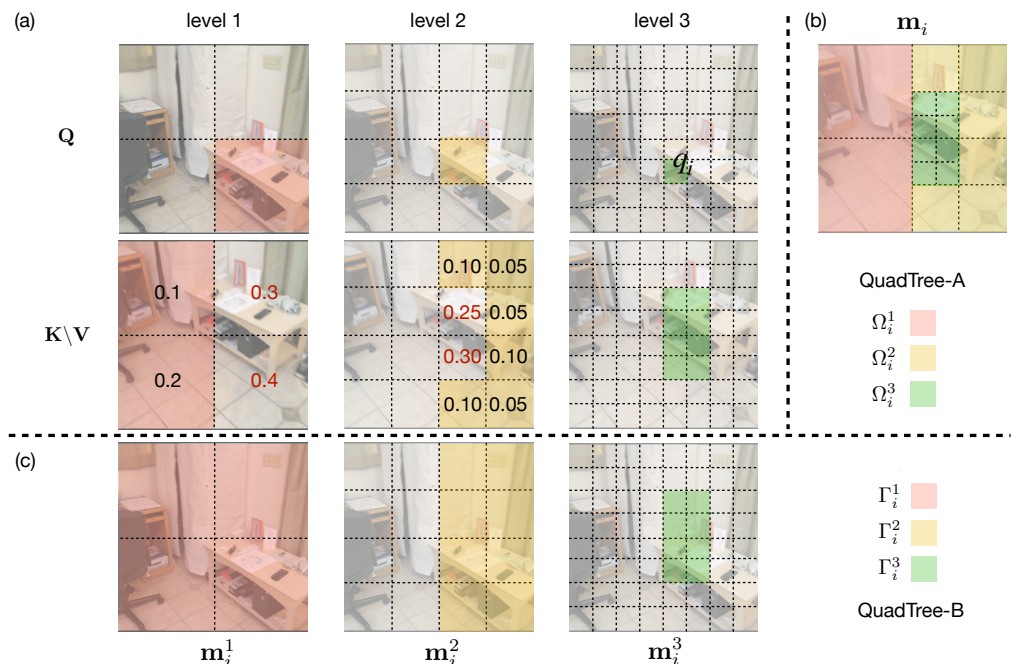

Figure 2: Illustration of quadtree message aggregation for a query token $q_i$. (a) shows the token pyramids and involved key/value tokens in each level. Attention scores are marked in the first two levels for clarification, and the top $K$ scores are highlighted in red. (b) shows message aggregation for QuadTree-A architecture. The message is assembled from different levels along a quadtree. (c) shows message aggregation for QuadTree-B architecture. The message is collected from overlapping regions from different levels.

following equation,

$$\mathbf{Q} = \mathbf{W}_q \mathbf{X}_1,$$
$$\mathbf{K} = \mathbf{W}_k \mathbf{X}_2,$$
$$\mathbf{V} = \mathbf{W}_v \mathbf{X}_2,$$

where $\mathbf{W}_q$, $\mathbf{W}_k$ and $\mathbf{W}_v$ are learnable parameters. Then, it performs message aggregation by computing the attention scores between query and key as following,

$$\mathbf{Y} = \text{softmax}(\frac{\mathbf{Q}\mathbf{K}^T}{\sqrt{C}})\mathbf{V}, \tag{1}$$

where $C$ is the embedding channel dimension. The above process has $O(N^2)$ computational complexity, where $N$ is the number of image patches in a vision transformer. This quadratic complexity hinders transformers from being applied to tasks requiring high resolution output. To address this problem, PVT (Wang et al. (2021c)) downsamples $\mathbf{K}$ and $\mathbf{V}$, while Swin Transformer (Liu et al. (2021)) limits the attention computation within local windows.

## 3.2 QUADTREE ATTENTION

In order to reduce the computational cost of vision transformers, we present QuadTree Attention. As the name implies, we borrow the idea from quadtrees, which are often used to partition a two-dimensional space by recursively subdividing it into four quadrants or regions. Quadtree attention computes attention in a coarse to fine manner. According to the results at the coarse level, irrelevant image regions are skipped quickly at the fine level. This design achieves less information loss while keeping high efficiency.

The same as the regular transformers, we first linearly project $\mathbf{X}_1$ and $\mathbf{X}_2$ to the query, key, and value tokens. To facilitate fast attention computation, we construct $L$-level pyramids for query $\mathbf{Q}$, key

**K**, and value **V** tokens by downsampling feature maps. For query and key tokens, we use average pooling layers. For value tokens, average pooling is used for cross attention tasks and convolutional-normalization-activation layers with stride 2 are used for self attention tasks if no special statement. As shown in Figure 1, after computing attention scores in the coarse level, for each query token, we select the top $K$ key tokens with the highest attention scores. At the fine level, query sub-tokens only need to be evaluated with those key sub-tokens that correspond to one of the selected $K$ key tokens at the coarse level. This process is repeated until the finest level. After computing the attention scores, we aggregate messages at all levels, where we design two architectures named as **QuadTree-A** and **QuadTree-B**.

**QuadTree-A**. Considering the $i$-th query token $\mathbf{q}_i$ at the finest level, we need to compute its received message $\mathbf{m}_i$ from all key tokens. This design assembles the full message by collecting partial messages from different pyramid levels. Specifically,

$$\mathbf{m}_i = \sum_{1 \le l \le L} \mathbf{m}_i^l, \tag{2}$$

where $\mathbf{m}_i^l$ indicates the partial message evaluated at level $l$. This partial message $\mathbf{m}_i^l$ assemble messages at the $l$-th level from tokens within the region $\Omega_i^l$, which will be defined later. In this way, messages from less related regions are computed from coarse levels, while messages from highly related regions are computed in fine levels. This scheme is illustrated in Figure 2 (b), message $\mathbf{m}_i$ is generated by assembling three partial messages that are computed from different image regions with different colors, which collectively cover the entire image space. The green region indicates the most relevant region and is evaluated at the finest level, while the red region is the most irrelevant region and is evaluated at the coarsest level. The region $\Omega_i^l$ can be defined as $\Gamma_i^l - \Gamma_i^{l+1}$, where the image region $\Gamma_i^l$ corresponds to the top $K$ tokens at the level $l - 1$. The regions $\Gamma_i^l$ are illustrated in Figure 2 (c). The region $\Gamma_i^1$ covers the entire image.

The partial messages are computed as,

$$\mathbf{m}_i^l = \sum_{j \in \Omega_i^l} s_{ij}^l \mathbf{v}_j^l, \tag{3}$$

where $s_{ij}^l$ is the attention score between the query and key tokens at level $l$. Figure 2 (a) highlights query and key tokens involved in computing $\mathbf{m}_i^l$ with the same color as $\Omega_i^l$. Attention scores are computed recursively,

$$s_{ij}^l = s_{ij}^{l-1} t_{ij}^l. \tag{4}$$

Here, $s_{ij}^{l-1}$ is the score of corresponding parent query and key tokens and $s_{ij}^1 = 1$. The tentative attention score $t_{ij}^l$ is evaluated according to Equation 1 among the $2 \times 2$ tokens of the same parent query token. For QuadTree-A, we use average pooling layers to downsample all query, key and value tokens.

**QuadTree-B**. The attention scores $s_{ij}^l$ in **QuadTree-A** are recursively computed from all levels, which makes scores smaller at finer levels and reduces the contributions of fine image features. Besides, fine level scores are also largely affected by the inaccuracy at coarse levels. So we design a different scheme, referred as **QuadTree-B** in this paper, to address this problem. Specifically, we compute $\mathbf{m}_i$ as a weighted average of the partial messages from different levels,

$$\mathbf{m}_i = \sum_{1 \le l \le L} w_i^l \mathbf{m}_i^l, \tag{5}$$

where $w_i^l$ is a learned weight. As shown in Figure 2 (c), the partial messages here overlap with each other, which are computed as,

$$\mathbf{m}_i^l = \text{Attention}(\mathbf{q}_i^l, \mathbf{K}_{\Gamma_i^l}^l, \mathbf{V}_{\Gamma_i^l}^l), \tag{6}$$

where Attention is the attention message computation as Equation 1. Here, $\mathbf{K}_{\Gamma_i^l}^l$ and $\mathbf{V}_{\Gamma_i^l}^l$ are matrices formed by stacking all keys and values within the region $\Gamma_i^l$.

Both **QuadTree-A** and **QuadTree-B** involve only sparse attention evaluation. Thus, our method largely reduces computational complexity. As analyzed in Appendix A.1, the computational complexity of our quadtree attention is linear to the number of tokens.

| | | AUC@5° | AUC@10° | AUC@20° |
|---|---|---|---|---|
| Others | ContextDesc + SGMNet(Chen et al. (2021)) | 15.4 | 32.3 | 48.8 |
| | SuperPoint + OANet (Zhang et al. (2019b)) | 11.8 | 26.9 | 43.9 |
| | SuperPoint + SuperGlue (Sarlin et al. (2020)) | **16.2** | **33.8** | **51.9** |
| | DRC-Net (Li et al. (2020)) | 7.7 | 17.9 | 30.5 |
| LoFTR-lite | Linear Att. (LoFTR) (Katharopoulos et al. (2020)) | 16.1 | 32.6 | 49.0 |
| | PVT (Wang et al., 2021c) | 16.2 | 32.7 | 49.2 |
| | QuadTree-A (ours, $K=8$) | 16.8 | 33.4 | 50.5 |
| | QuadTree-B (ours, $K=8$) | **17.4** | **34.4** | **51.6** |
| LoFTR | Linear Att. (LoFTR) $\star$ (Sun et al. (2021), 64 GPUs) | 22.1 | 40.8 | 57.6 |
| | Linear Att. (LoFTR) (Katharopoulos et al. (2020)) | 21.1 | 39.5 | 56.6 |
| | QuadTree-B (ours, $K=8$) | 23.0 | 41.7 | 58.5 |
| | QuadTree-B$*$ (ours, $K=16$) | **24.9** | **44.7** | **61.6** |

Table 1: Results on feature matching. The symbol $\star$ indicates results cited from (Sun et al., 2021), where the model is trained with a batch size of 64 on 64 GPUs (a more preferable setting than ours). The symbol $*$ indicates we use the ViT (Dosovitskiy et al., 2020)-like architecture for transformer blocks. For PVT and our method, we replace the original linear attention in LoFTR with corresponding attentions.

**Multiscale position encoding**. The computation of attention is permutation invariant to tokens, and thus positional information is missed. To address this problem, we adopt the locally-enhanced positional encoding (LePE) (Dong et al., 2021) at each level to design a multiscale position encoding. Specifically, for level $l$, we apply unshared depth-wise convolution layers to value tokens $\mathbf{V}^l$ to encode the positional information.

## 4 EXPERIMENT

We experiment our quadtree transformer with four representative tasks, including feature matching, stereo, image classification, and object detection. The first two tasks require cross attention to fuse information across different images, while the latter two involve only self-attention. We implement our quadtree transformer using PyTorch and CUDA kernels. More implementation details are provided in Appendix B.

### 4.1 CROSS ATTENTION TASKS

#### 4.1.1 FEATURE MATCHING

Finding feature correspondence (Luo et al., 2019; DeTone et al., 2018) across different images is a precedent problem for many 3D computer vision tasks. It is typically evaluated by the accuracy of the camera pose estimated from the corresponding points. We follow the framework proposed in a recent state-of-the-art work LoFTR (Sun et al., 2021), which consists of a CNN-based feature extractor and a transformer-based matcher. We replace the linear transformer (Katharopoulos et al., 2020) in LoFTR with our quadtree transformer. Besides, we also implement a new version of LoFTR with the spatial reduction (SR) attention (Wang et al., 2021c) for additional comparison.

**Setting.** We experiment on ScanNet (Dai et al., 2017) with 1,513 scans. In order to accelerate training, we design the LoFTR-lite setting, which uses half of the feature channels of LoFTR and 453 training scans. Ablation studies in section 4.3 are conducted in this setting. We train both LoFTR-lite and LoFTR for 30 epochs with batch size 8. For quadtree transformer, we build pyramids of three levels with the coarsest resolution at $15 \times 20$ pixels. We set the parameter $K$ to 8 at the finest level, and double it at coarser levels. For the SR attention, we average pool the value and key tokens to the size $8 \times 8$ to keep similar memory usage and flops as our quadtree attention. More details are included in Appendix B.1.

**Results.** Table 1 shows the AUC of camera pose errors[1] under (5°, 10°, 20°). We can see that the SR attention achieves similar results with linear transformer. In comparison, both QuadTree-A and QuadTree-B outperform linear transformer and SR attention by a large margin. Quadtree-B generally performs better than Quadtree-A. Quadtree-B has 2.6 and 1.9 improvements in terms of AUC@20° over linear transformer on LoFTR-lite and LoFTR respectively. To further enhance the

---

[1]Camera pose errors are evaluated as the differences between estimated and ground truth camera orientation and translation direction, both measured in degrees.

|  | EPE (px) | IOU | Flops (G) | Mem. (MB) |
|---|---|---|---|---|
| GA-Net (Zhang et al., 2019a) | 0.89 | / | / | / |
| GWC-Net (Guo et al., 2019) | 0.97 | / | 305 | 4339 |
| Bi3D (Badki et al., 2020) | 1.16 | / | 897 | 10031 |
| STTR (Vanilla Transformer) (Li et al., 2021) | **0.45** | 0.92 | 490 | 8507 |
| QuadTree-B (ours, $K = 6$) | 0.46 | **0.99** | **254 (52%)** | **5381 (63%)** |

Table 2: Results of stereo matching. QuadTree-B achieves similar performance as STTR but with significantly lower flops and memory usage.

results, we train a model with $K = 16$ and leverage a ViT (Dosovitskiy et al., 2020)-like transformer archtecture instead of the original one used in (Sun et al., 2021). This model achieves 4 improvements on AUC@20° over (Sun et al., 2021), where the LoFTR model is trained with a batch size of 64 with 64 GPUs, a more preferable setting leading to slightly better results than our linear transformer implementation shown in Table 1.

### 4.1.2 STEREO MATCHING

Stereo matching aims to find corresponding pixels on epipolar lines between two rectified images. The recent work STTR (Li et al., 2021) applies transformers to feature points between epipolar lines and achieves state-of-the-art performance. Note here, both self- and cross attention are applied along epipolar lines, pixels across different lines are not considered in the attention computation. We replace the standard transformer in STTR (Li et al., 2021) with our quadtree transformer.

**Setting.** We experiment on the Scene Flow FlyingThings3D (Mayer et al., 2016) synthetic dataset, which contains 25,466 images with a resolution of $960 \times 540$. We build pyramids of four levels to evaluate quadtree attention. While the STTR is applied to features of 1/3 of image resolution, we use feature maps of 1/2 of image resolution. More details about the network are included in Appendix B.2.

**Results.** We report EPE (End-Point-Error) in non-occluded regions and IOU (Intersection-over-Union) for occlusion estimation in Table 2 as (Li et al., 2021). Computational complexity and memory usage are also reported. Compared with STTR based on the standard transformer, our quadtree transformer achieves similar EPE (0.45 px vs 0.46 px) and higher IOU for occlusion estimation, but with much lower computational and memory costs, with only 52% FLOPs and 63% memory consumption.

## 4.2 SELF-ATTENTION TASK

This section presents results on image classification and object detection. In the past, convolutional neural networks (CNNs) have dominated these tasks for a long time. Recently, vision transformers (Dosovitskiy et al., 2020; Liu et al., 2021; Wang et al., 2021c) show excellent potential on these problems, thanks to their capability in capturing long-range interactions. To compare our method with these vision transformers on image classification, we use the public codes of PVTv2 (Wang et al., 2021c) and replace all the spatial reduction attention with our quadtree attention. For object detection, we further apply a representative object detection framework, RetinaNet (Lin et al., 2017), which is a widely used single-stage object detector.

### 4.2.1 IMAGE CLASSIFICATION

**Settings.** We evaluate image classification on the ImageNet-1K dataset (Deng et al., 2009), which consists of 1.28M training images and 50K validation images from 1,000 categories. We build token pyramids with the coarsest level at a resolution of $7 \times 7$ and set $K = 8$. We crop and resize the input images to $224 \times 224$ pixels and train the model with a mini-batch of 128. All models are trained for 300 epochs from scratch on 8 GPUs. All the other training settings are the same as in (Wang et al., 2021c). We build five different quadtree transformers at different complexity, named as b0, b1, b2, b3, b4. These models are gradually deeper and wider. More configuration details can be found in Appendix B.3.

**Results.** We provide the top-1 accuracy of various methods and network settings in Table 3. These results are grouped into five sections, each with several methods of similar network complexity, as

| | Param (M) | Flops (G) | Top1 (%) |
|---|---|---|---|
| PVTv2-b0 (Wang et al., 2021b) | 3.7 | 0.6 | 70.5 |
| QuadTree-A-b0 (ours) | **3.4** | 0.6 | 70.9 |
| QuadTree-B-b0 (ours) | 3.5 | 0.7 | **72.0** |
| ResNet18 (He et al., 2016) | **11.7** | **1.8** | 69.8 |
| PVTv1-Tiny (Wang et al., 2021c) | 13.2 | 2.1 | 75.1 |
| PVTv2-b1 (Wang et al., 2021b) | 14.0 | 2.1 | 78.7 |
| QuadTree-B-b1 (ours) | 13.6 | 2.3 | **80.0** |
| ResNet50 (He et al., 2016) | 25.1 | 4.1 | 76.4 |
| ResNeXt50-32x4d (Xie et al., 2017) | 25.0 | 4.3 | 77.6 |
| RegNetY-4G (Radosavovic et al., 2020) | 21.0 | 4.0 | 80.0 |
| DeiT-Small/16 (Touvron et al., 2021) | 22.1 | 4.6 | 79.9 |
| Swin-T (Liu et al., 2021) | 29.0 | 4.5 | 81.3 |
| TNT-S (Han et al., 2021) | 23.8 | 5.2 | 81.3 |
| CeiT (Yuan et al., 2021a) | 24.2 | 4.5 | 82.0 |
| PVTv2-b2 (Wang et al., 2021c) | 25.4 | **4.0** | 82.0 |
| Focal-T (Yang et al., 2021) | 29.1 | 4.9 | 82.2 |
| QuadTree-B-b2 (ours) | 24.2 | 4.5 | **82.7** |
| ResNet101 (He et al., 2016) | 44.7 | 7.9 | 77.4 |
| ResNeXt101-32x4d (Xie et al., 2017) | 44.2 | 8.0 | 78.8 |
| RegNetY-8G (Radosavovic et al., 2020) | 39.0 | 8.0 | 81.7 |
| CvT-21 (Wu et al., 2021) | **32.0** | 7.1 | 82.5 |
| PVTv2-b3 (Wang et al., 2021c) | 45.2 | **6.9** | 83.2 |
| Quadtree-B-b3 (ours) | 46.3 | 7.8 | **83.7** |
| ResNet152 (He et al., 2016) | 60.2 | 11.6 | 78.3 |
| T2T-ViTt-24 (Yuan et al., 2021b) | 64.0 | 15.0 | 82.2 |
| Swin-S (Liu et al., 2021) | **50.0** | **8.7** | 83.0 |
| Focal-Small (Yang et al., 2021) | 51.1 | 9.1 | 83.5 |
| PVTv2-b4 (Wang et al., 2021c) | 62.6 | 10.1 | 83.6 |
| Quadtree-B-b4 (ours) | 64.2 | 11.5 | **84.0** |

Table 3: Image classification results. We report top-1 accuracy on the ImageNet validation set.

| | Flops (G) | AP | $AP_{50}$ | $AP_{75}$ | $AP_S$ | $AP_M$ | $AP_L$ |
|---|---|---|---|---|---|---|---|
| PVTv2-b0 (Wang et al., 2021b) | 28.3 | 37.2 | 57.2 | 39.5 | 23.1 | 40.4 | 49.7 |
| QuadTree-A-b0 (K=32, ours) | **16.0** | 37.0 | 56.8 | 38.9 | **22.8** | 39.7 | 50.0 |
| QuadTree-B-b0 (K=32, ours) | 16.5 | **38.4** | **58.7** | **41.1** | 22.5 | **41.7** | **51.6** |
| ResNet18 (He et al., 2016) | 38.6 | 31.8 | 49.6 | 33.6 | 16.3 | 34.3 | 43.2 |
| PVTv1-Tiny (Wang et al., 2021c) | 72.5 | 36.7 | 56.9 | 38.9 | 22.6 | 38.8 | 50.7 |
| PVTv2-b1 (Wang et al., 2021b) | 78.8 | 41.2 | 61.9 | 43.9 | 25.4 | 44.5 | 54.3 |
| Quadtree-B-b1 (K=32, ours) | **56.2** | **42.6** | **63.6** | **45.3** | **26.8** | **46.1** | **57.2** |
| ResNet50 (He et al., 2016) | 87.3 | 36.3 | 55.3 | 38.6 | 19.3 | 40.0 | 48.8 |
| ResNet101 (He et al., 2016) | 166.3 | 38.5 | 57.8 | 41.2 | 21.4 | 42.6 | 51.1 |
| ResNeXt101-32x4d (Xie et al., 2017) | 170.2 | 39.9 | 59.6 | 42.7 | 22.3 | 44.2 | 52.5 |
| PVTv1-small (Wang et al., 2021c) | 139.8 | 36.7 | 56.9 | 38.9 | 25.0 | 42.9 | 55.7 |
| PVTv2-b2 (Wang et al., 2021c) | 149.1 | 44.6 | 65.6 | 47.6 | 27.4 | 48.8 | 58.6 |
| QuadTree-B-b2 (K=32, ours) | **108.6** | **46.2** | **67.2** | **49.5** | **29.0** | **50.1** | **61.8** |
| PVTv1-Medium (Wang et al., 2021c) | 237.4 | 41.9 | 63.1 | 44.3 | 25.0 | 44.9 | 57.6 |
| PVTv2-b3 (Wang et al., 2021b) | 243.0 | 45.9 | 66.8 | 49.3 | 28.6 | 49.8 | 61.4 |
| QuadTree-B-b3 (ours) | **193.9** | **47.3** | **68.2** | **50.6** | **30.4** | **51.3** | **62.9** |
| PVTv1-Large (Wang et al., 2021c) | 346.6 | 42.6 | 63.7 | 45.4 | 25.8 | 46.0 | 58.4 |
| PVTv2-b4 (Wang et al., 2021b) | 353.3 | 46.1 | 66.9 | 49.2 | 28.4 | 50.0 | 62.2 |
| QuadTree-B-b4 (ours) | **283.9** | **47.9** | **69.1** | **51.3** | **29.4** | **52.2** | **63.9** |

Table 4: Object detection results on COCO val2017 with RetinaNet. We use PVTv2 backbone and replace the reduction attention with quadtree attention. 'Flops' is the backbone flops for input image size of $800 \times 1,333$.

indicated by the number of parameters. As shown in Table 3, QuadTree-B outperforms PVTv2 by 0.4%-1.5% in top-1 accuracy with fewer parameters. Swin Transformer-S adopts local attention and is surpassed by our QuadTree-B-b2 by 1.0% in top-1 accuracy. This result proves that global information is important. In general, our quadtree transformer leverages both global information at the coarse level and local information at fine levels, and outperforms both PVTv2 and Swin Transformer.

| | ImageNet-1K | | COCO (RetinaNet) | | | |
|---|---|---|---|---|---|---|
| | Flops (G) | Top-1 (%) | Mem. (MB) | AP | $AP_{50}$ | $AP_{75}$ |
| PVTv2 (Wang et al., 2021b) | **0.6** | 70.5 | 574 | 37.2 | 57.2 | 39.5 |
| PVTv2+LePE (Dong et al., 2021) | **0.6** | 70.9 | 574 | 37.6 | 57.8 | 39.9 |
| Swin (Liu et al., 2021) | **0.6** | 70.5 | **308** | 35.3 | 54.2 | 37.4 |
| Swin+LePE | 0.6 | 70.7 | **308** | 35.8 | 55.3 | 37.7 |
| Focal Attention (Yang et al., 2021) | 0.7 | 71.6 | 732 | 37.5 | 57.6 | 39.5 |
| Focal Attention+LePE | 0.7 | 71.5 | 732 | 37.1 | 57.0 | 39.4 |
| QuadTree-B | **0.6** | **72.0** | 339 | **38.4** | **58.8** | **41.1** |

Table 5: To fairly compare with Swin, PVT, Focal attention and our method, we replace the attention module in PVTv2-b0 with different types of attention and same position encoding method LePE and run image classification and object detection respectively.

### 4.2.2 OBJECT DETECTION

**Settings.** We experiment on the COCO dataset. All models are trained on COCO train 2017 (118k images) and evaluated on val 2017 (5k images). We initialize the quadtree backbone with the weights pre-trained on ImageNet. We adopt the same setting as PVTv2, training the model with a batch size of 16 and AdamW optimizer with an initial learning rate of $1 \times 10^{-4}$ for 12 epochs. We use the standard metric average precision to evaluate our method.

**Results.** We mainly compare our method with PVTv2, ResNet (He et al., 2016), and ResNeXt (Xie et al., 2017) using detection framework of RetinaNet (Lin et al., 2017), which are state-of-the-art backbones for dense prediction. Table 4 lists the average precision of different methods and their backbone flops for images of resolution of $800 \times 1,333$. Benefiting from the coarse to fine mechanism, a small $K$ is enough for our method. Thus, the computation can be reduced when using high resolution images. We can see that QuadTree-B achieves higher performance, but with much fewer flops than PVTv2. Our quadtree transformer also outperforms ResNet and ResNeXt. For example, QuadTree-B-b2 outperform ResNet101 and ResNeXt101-32x4d by 7.7 AP and 6.3 AP respectively with about 40% backbone flops reduction. We also show Mask-RCNN results (He et al., 2017) in Appendix. E.

### 4.3 COMPARISON WITH OTHER ATTENTION MECHANISMS

For a fair comparison with other attention mechanisms, we test these attention mechanisms under the same backbone and training settings. Specifically, we replace the original attention module in PVTv2-b0 with the attention method used in Swin Transformer and Focal Transformer. For more fair comparison, we adopt the same positional encoding LePE (Dong et al., 2021) to PVTv2, Swin and Focal transformer. As shown in Table 5, QuadTree attention obtain consistently better performance than Swin and PVTv2 in both classification task and detection task. Compared with focal attention, our method gets 0.9 higher AP in object detection, which might be because that QuadTree attention can always cover the whole images, while Focal attention only covers $1/6$ of the image in the first stage. More experiments on Swin-like architecture can be found in Appendix E.

For cross attention tasks, we also provide visualization of attention score as shown in Fig.5 in Appendix E. Our method can attend to much more related regions than PVT (Wang et al., 2021b) and Linear attention (Katharopoulos et al., 2020).

## 5 CONCLUSION

We introduce QuadTree Attention to reduce the computational complexity of vision transformers from quadratic to linear. Quadtree transformers build token pyramids and compute attention in a coarse-to-fine manner. At each level, top $K$ regions with the highest attention scores are selected, such that in finer level, computation in irrelevant regions can be quickly skipped. Quadtree attention can be applied to cross attention as well as self-attention. It achieves state-of-the-art performance in various tasks including feature matching, stereo, image classification, and object detection.

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

## A  Appendix

### A.1  Complexity analysis

In this section, we analyze the computational complexity of quadtree attention. Suppose the lengths of the query tokens, key tokens, and value tokens are all $H \times W$. We build token pyramids of $L$ levels, the $l^{th}$ level has a token length of $\frac{HW}{4^{l-1}}$. The flops of computing quadtree attention is,

$$\text{Flops} = 2(H_0^2 W_0^2 + \sum_{l-2}^{L-1} \frac{4KHW}{4^{l-1}})$$

$$= 2(H_0^2 W_0^2 + \frac{4}{3}(1 - 4^{1-L})KHW).$$

Here, $H_0$ and $W_0$ are the height and width of the coarsest level of token pyramids. Therefore, $H_0^2 W_0^2$ is a constant and the computational complexity is $O(KHW)$. Since $K$ is a constant number, the complexity of quadtree attention is linear to the number of tokens.

|  | AUC@5° | AUC@10° | AUC@20° |
|---|---|---|---|
| DRC-Net (Li et al., 2020) | 27.0 | 43.0 | 58.3 |
| SuperPoint + SuperGlue (Sarlin et al., 2020) | 42.2 | 61.2 | 76.0 |
| LoFTR (Sun et al., 2021) | 52.8 | 69.2 | 81.2 |
| QuadTree-B (ours, K=16) | **54.6** | **70.5** | **82.2** |

Table 6: Feature matching results on megadepth. Our method obtains better performance than other methods.

## B  ADDITIONAL EXPERIMENTS AND IMPLEMENTATION DETAILS

### B.1  FEATURE MATCHING

**Implementation details.** We train and evaluate the model in ScanNet (Dai et al., 2017), where 230M image pairs is sampled for training, with overlapping scores between 0.4 and 0.8. ScanNet provides RGB images, depth maps, and ground truth camera poses on a well-defined training and testing split. Following the same evaluation settings as Sarlin et al. (2020) and Sun et al. (2021), we evaluate our method on the 1,500 testing pairs from (Sarlin et al., 2020). For both trainig and testing, all images and depth maps are resized to $640 \times 480$. Following (Sun et al., 2021), we compute the camera pose by solving the essential matrix from predicted matches with RANSAC. We report the AUC of the pose error at thresholds ($5°$, $10°$, $20°$), where the pose error is defined as the maximum of angular error in rotation and translation. We only replace the coarse level transformer with quadtree attention.

**Results of megadepth.** We show our results on Megadepth (Li & Snavely, 2018) in Table 6. We can see our method outperforms others by a large margin.

### B.2  STEREO MATCHING

Our network is based on the STTR (Li et al., 2021), where we replace the standard transformer with our quadtree transformer. The network consists of a CNN backbone which outputs feature maps of 1/2 image resolution, a quadtree transformer with both self- and cross attention, a regression head with optimal transport layers (Cuturi, 2013), and a context adjust layer to refine the disparity. Six self- and cross attention layers are used with 128 channels. We build pyramids with four levels for quadtree attention, and apply the Sinkhorn algorithm (Cuturi, 2013) for 10 iteration for optimal transport. We follow STTR to train the network, with 15 epochs of AdamW optimizer. OneCycle learning rate scheduler is used with a leaning rate of 6e-4 and a batch size of 8.

### B.3  IMAGE CLASSIFICATION

This paragraph introduces the details of PVTv2-b0, b1, b2, b3, b4. All these five networks have 4 stages. Each stage is down-sampled from the previous stage by a stride of 2. The feature resolutions for each stage are $\frac{H}{4} \times \frac{W}{4}$, $\frac{H}{8} \times \frac{W}{8}$, $\frac{H}{16} \times \frac{W}{16}$ and $\frac{H}{32} \times \frac{W}{32}$ respectively, where $H$ and $W$ is the image height and width. For each stage, $M$ quadtree transformers are used with a channel number of $I$ and head number of $J$. For the network PVTv2-b0, the parameters $M$, $I$, $J$ are set to $[2, 2, 2, 2]$, $[32, 64, 160, 256]$, $[1, 2, 5, 8]$ at each stage respectively. For the network PVTv2-b1, the parameters $M$, $I$, $J$ are set to $[2, 2, 2, 2]$, $[64, 128, 320, 512]$, $[1, 2, 5, 8]$ respectively. For PVTv2-b2, the parameters $M$, $I$, $J$ are set to $[3, 4, 6, 3]$, $[64, 128, 320, 512]$, $[1, 2, 5, 8]$ respectively. For PVTv2-b3, the parameters $M$, $I$, $J$ are set to $[3, 4, 18, 3]$, $[64, 128, 320, 512]$, $[1, 2, 5, 8]$ respectively. For PVTv2-b4, the parameters $M$, $I$, $J$ are set to $[3, 8, 27, 3]$, $[64, 128, 320, 512]$, $[1, 2, 5, 8]$ respectively.

### B.4  OBJECT DETECTION AND INSTANCE SEGMENTATION

We show the object detection and instance segmentation results of Mask-RCNN (He et al., 2017) in Table 7 and Table 8 in different training settings. In Table 7, we train Mask-RCNN for 12 epoch and resize the image to $800 \times 1333$ while In Table 8, we train the model for 36 epochs and resize the training images to different scales for data augmentation. We can see that the QuadTree attention obtains consistently better performance than other methods.

| | $AP^b$ | $AP^b_{50}$ | $AP^b_{75}$ | $AP^m$ | $AP^m_{50}$ | $AP^m_{75}$ |
|---|---|---|---|---|---|---|
| PVTv2-b0 (Wang et al., 2021b) | 38.2 | 60.5 | 40.7 | 36.2 | 57.8 | 38.6 |
| QuadTree-B-b0 (K=32, ours) | **38.8** | **60.7** | **42.1** | **36.5** | **58.0** | **39.1** |
| ResNet18 (He et al., 2016) | 34.0 | 54.0 | 36.7 | 31.2 | 51.0 | 32.7 |
| PVTv1-Tiny (Wang et al., 2021c) | 36.7 | 59.2 | 39.3 | 35.1 | 56.7 | 37.3 |
| PVTv2-b1 (Wang et al., 2021b) | 41.8 | 64.3 | 45.9 | 38.8 | 61.2 | 41.6 |
| Quadtree-B-b1 (K=32, ours) | **43.5** | **65.6** | **47.6** | **40.1** | **62.6** | **43.3** |
| ResNet50 (He et al., 2016) | 38.0 | 58.6 | 41.4 | 34.4 | 55.1 | 36.7 |
| ResNet101 (He et al., 2016) | 40.4 | 61.1 | 44.2 | 36.4 | 57.7 | 38.8 |
| ResNeXt101-32x4d (Xie et al., 2017) | 41.9 | 62.5 | 45.9 | 37.5 | 59.4 | 40.2 |
| PVTv1-small (Wang et al., 2021c) | 40.4 | 62.9 | 43.8 | 37.8 | 60.1 | 40.3 |
| PVTv2-b2 (Wang et al., 2021b) | 45.3 | 67.1 | 49.6 | 41.2 | 64.2 | 44.4 |
| QuadTree-B-b2 (K=32, ours) | **46.7** | **68.5** | **51.2** | **42.4** | **65.7** | **45.7** |
| PVTv1-Medium (Wang et al., 2021c) | 42.0 | 64.4 | 45.6 | 39.0 | 61.6 | 42.1 |
| PVTv2-b3 (Wang et al., 2021b) | 45.9 | 66.8 | 49.3 | 28.6 | 49.8 | 61.4 |
| QuadTree-B-b3 | **48.3** | **69.6** | **52.8** | **43.3** | **66.8** | **46.6** |
| PVTv1-Large (Wang et al., 2021c) | 42.9 | 65.0 | 46.6 | 39.5 | 61.9 | 42.5 |
| PVTv2-b4 (Wang et al., 2021b) | 47.5 | 68.7 | 52.0 | 42.7 | 66.1 | 46.1 |
| QuadTree-B-b4 | **48.6** | **69.5** | **53.3** | **43.6** | **66.9** | **47.4** |

Table 7: Object detection results on COCO val2017 with Mask-RCNN. We use PVTv2 backbone and replace the reduction attention with quadtree attention.

| | #Params | AP | $AP_{50}$ | $AP_{75}$ | $AP_S$ | $AP_M$ | $AP_L$ |
|---|---|---|---|---|---|---|---|
| QuadTree-B-b0 | **23.4** | **42.4** | **64.5** | **45.9** | **38.9** | **61.6** | **41.6** |
| QuadTree-B-b1 | **33.3** | **46.4** | **68.6** | **50.7** | **41.9** | **65.6** | **44.7** |
| Swin-T (Liu et al., 2021) | 47.8 | 46.0 | 68.1 | 50.3 | 41.6 | 65.1 | 44.9 |
| Focal-T (Yang et al., 2021) | 48.8 | 47.2 | 69.4 | 51.9 | 42.7 | 66.5 | 45.9 |
| QuadTree-B-b2 | **44.8** | **49.3** | **70.7** | **53.9** | **43.9** | **67.6** | **47.4** |
| Swin-S (Liu et al., 2021) | **69.1** | 48.5 | 70.2 | 53.5 | 43.3 | 67.3 | 46.6 |
| Focal-S Yang et al. (2021) | 71.2 | 48.8 | **70.5** | 53.6 | 43.8 | 67.7 | 47.2 |
| QuadTree-B-b3 | 70.0 | **49.6** | 70.4 | **54.2** | **44.0** | 67.7 | 47.5 |

Table 8: Object detection results on COCO val2017 with Mask-RCNN training with 36 epochs and multi-scale data argumentation strategy. We use PVTv2 backbone and replace the reduction attention with quadtree attention.

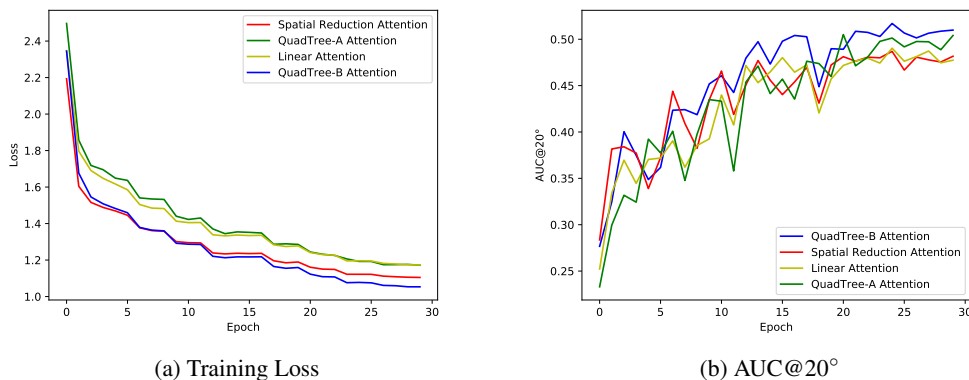

(a) Training Loss          (b) AUC@20°

Figure 3: Loss and AUC@20° of image matching.

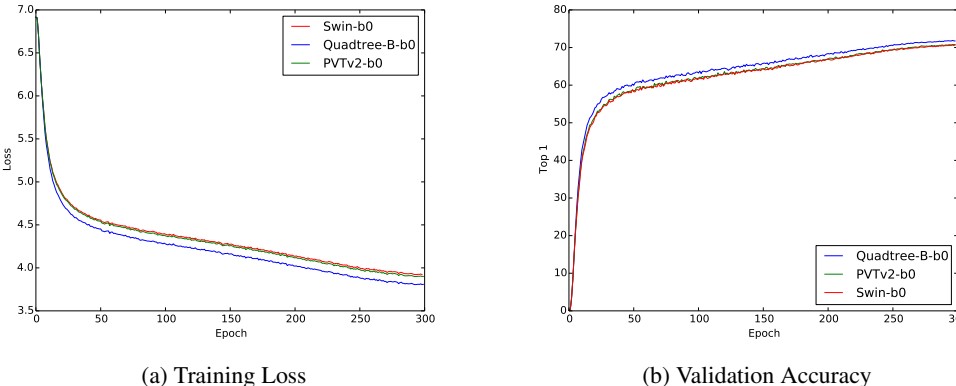

(a) Training Loss                                                   (b) Validation Accuracy

Figure 4: Loss and top 1 accuracy of image classification for PVTv2-b0 archtecture.

| | ImageNet | | | COCO (RetinaNet) | | |
|---|---|---|---|---|---|---|
| | Param. (M) | Flops (G) | Top1 (%) | AP | $AP_{50}$ | $AP_{75}$ |
| Swin-T (Liu et al., 2021) | **29** | **4.5** | 81.3 | 42.0 | \ | \ |
| Focal-T (Yang et al., 2021) | **29** | 4.9 | **82.2** | 43.7 | \ | \ |
| Quadtree-B | 30 | 4.6 | **82.2** | **44.6** | **65.8** | **47.7** |

Table 9: Comparison under Swin-T settings in image classification and object detection.

## C  TRAINING LOSS

**Feature matching.** We plot the training loss and validation performance for LoFTR-lite in Figure 3 for different efficient transformers, including spatial reduction (SR) transformer (Wang et al., 2021c), linear transformer (Katharopoulos et al., 2020), our Quadtree-A, and Quadtree-B transformers. We can see quadtree-B transformer obtains consistently lower training loss and higher performance over other three transformers. In addition, it is also noted that the spatial reduction (SR) transformer has lower training but worse AUC@20° than QuadTree-A attention, which indicates that it cannot generalize well.

**Image classification**. We also show traning and validation curve for image classification task with respective to different attentions in Fig. 4. Compared with Swin Transformer (Liu et al., 2021) and PVT (Wang et al., 2021c), the loss of Quadtree attention is consistently lower and the top 1 accuracy is higher.

## D  RUNNING TIME

Currently, we only implement a naive CUDA kernel without many optimizations and it is not as efficient as the well-optimized dense GPU matrix operation. We test the running time of Retinanet under PVTv2-b0 architecture. For PVTv2-b0, The running time is 0.026s to forward one image and for Quadtree-b0, the running time is 0.046s for forwarding once. However, Quadtree-b0 has much lower memory usage than PVTv2-b0. Quadtree-b0 consumes about 339MB while PVTv2-b0 consumes about 574MB for one $800 \times 1333$ image.

## E  ABLATIONS

**QuadTree-A vs QuadTree-B**. QuadTree-B architecture consistently outperforms QuadTree-A in feature matching, image classification, and detection experiments. We analyze its reason as shown in Figure 5, where (d) and (e) show the attention score maps of QuadTree-A and QuadTree-B at different levels for the same point in the query image shown in (a). It is clear that the QuadTree-B has more accurate score maps, and is less affected by the inaccuracy in coarse level score estimation. We further visualize the attention scores of spatial reduction (SR) attention (Wang et al., 2021c) and

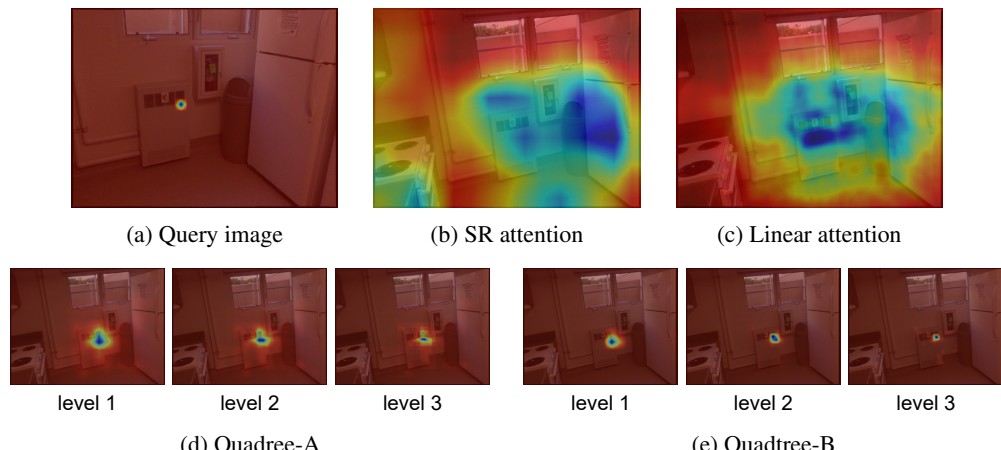

Figure 5: Score map visualization of different attention methods for one patch in the query image. The first row shows score maps of spatial reduction attention and linear attention. The second row shows score maps of QuadTree-A and QuadTree-B at different levels, and the left image is the coarsest level, while the right image is the finest level for both sub-figures.

| | ImageNet | | COCO (RetinaNet) | | |
|---|---|---|---|---|---|
| | Flops (G) | Top 1(%) | AP | $AP_{50}$ | $AP_{75}$ |
| Quadtree-B-b2 | 4.3 | 82.6 | 44.9 | 66.2 | 47.7 |
| Quadtree-B-b2+MPE | 4.3 | **82.7** | **46.2** | **67.2** | **49.5** |

Table 10: Ablation on multiscale position encoding.

linear transformer (Katharopoulos et al., 2020) in (b) and (c). We can see that SR attention and linear transformer attend the query token on large unrelated regions due to the loss of fine-grained information. In contrast, our quadtree transformer focus on the most relevant area.

**Comparison with Swin Transformer and Focal Transformer.** We compare with Swin Transformer and Focal Transformer in Table. 9 using the released codes. We replace the corresponding attention in Swin Transformer with Quadtree-B attention. Our method obtains 0.9% higher top 1 accuracy than Swin Transformer and 2.6% higher AP in object detection. Compared with Focal transformer, quadtree attention achieve the same top 1 accuracy in classification with fewer flops, and 0.9% higher AP in object detection.

**Multiscale position encoding.** We compare our method with or without multiscale position encoding (MPE). For Quadtree-B-b2 model, MPE can bring an improvement of 1.3 on object detection.

**Top $K$ numbers.** Table 11 and Table 12 shows the performance of QuadTree-B architecture with different value of $K$ for object detection and feature matching respectively. The performance is improved when $K$ becomes larger and saturates quickly. This indicates only a few tokens with high attention scores should be subdivided in the next level for computing attentions.

| | AP | $AP_{50}$ | $AP_{75}$ |
|---|---|---|---|
| $K = 1$ | 37.3 | 57.2 | 39.4 |
| $K = 8$ | 38.0 | 58.2 | 40.4 |
| $K = 16$ | 38.4 | 58.7 | 41.1 |
| $K = 32$ | **38.5** | **58.8** | **41.1** |

Table 11: The performance of QuadTree-B under different $K$ in object detection.

| | AUC@5° | AUC@10° | AUC@20° |
|---|---|---|---|
| $K = 1$ | 15.7 | 32.3 | 48.9 |
| $K = 4$ | 16.2 | 33.3 | 50.8 |
| $K = 8$ | 17.4 | 34.4 | 51.6 |
| $K = 16$ | **17.7** | **34.6** | **51.7** |

Table 12: The performance of QuadTree-B under different $K$ in feature matching

