# OpenReview forum: "Quadtree Attention for Vision Transformers"
_ICLR.cc/2022/Conference — ICLR 2022 Poster_

### Official Review · Reviewer_fyBu · 2021-11-01

**Correctness:** 4
**Technical Novelty And Significance:** 3
**Empirical Novelty And Significance:** 3
**Recommendation:** 6
**Confidence:** 4

**Main Review:**

Strength:
+ Introducing the quadtree to the vision transformers is interesting and makes sense. It can efficiently achieve long-range dependence while keeping the local details.
+ The paper is overall well-written and well-organized.
+ The reported performance is competitive against many state-of-the-art vision transformers.

Weakness:
- Top-k assignment for tokens in the quadtree is nondifferentiable, which may reduce the generalization. How does the proposed method achieve end-to-end training?
- The quadtree aggregates features in an unstructured and iterative way. It is unfriendly to the parallel devices and probably takes more latency than the dense attention. It would be nice to provide the actual throughput on GPUs to demonstrate the efficiency.

**Summary Of The Paper:**

The paper proposes an efficient attention algorithm based on quadtree for vision transformers. It establishes a feature pyramid for tokens and aggregates sparse key-value features in each pyramid level. The method achieves competitive performance on the stereo, image classification and object detection.

**Summary Of The Review:**

The paper proposes a novel efficient vision transformer based on quadtrees, which is interesting and technical sound. It achieves competitive performance in various vision tasks.

---

> ### Author Response · Authors · 2021-11-16
> **Responses to R4**
>
> We appreciate for the constructive suggestions.
> 1. Top-k assignment for tokens in the quadtree is nondifferentiable
>
>    Although top K is not differentiable, the gradient can also be passed to the selected query, key, value tokens, which enables end-to-end training. Top K selection is a common practice in network design, such as [1,2].
>
> 2. Running time.
>
>    Currently, we only implement naive CUDA codes without many optimizations, and it is not as efficient as the well-optimized dense GPU matrix operation. We test the running time of Retinanet under PVTv2-b0 architecture. We test the speed under a batch size of 10. For PVTv2-b0, the running time is 0.022s to forward one image and for Quadtree-b0, the running time is 0.038s for forwarding one image on average. However, Quadtree-b0 has much lower memory usage than PVTv2-b0. Quadtree-b0 consumes about 339 MB memory, while PVTv2-b0 consumes about 574 MB memory for one $800\times 1333$ image. Further speed-up can be achieved by more engineering optimizations, such as {https://openreview.net/forum?id=agBJ7SYcUVb}.
>
> [1]: Kvt: k-nn attention for boosting vision transformers, arxiv 2021
>
> [2]: Learning Camera Localization via Dense Scene Matching, CVPR 2021

---

> ### Comment · Reviewer_fyBu · 2021-12-01
> **Final decision**
>
> The response addressed my major concerns. I lean to keep my initial rate.

---

### Official Review · Reviewer_PDon · 2021-11-01

**Correctness:** 3
**Technical Novelty And Significance:** 3
**Empirical Novelty And Significance:** 3
**Recommendation:** 8
**Confidence:** 4

**Main Review:**

Pros.

1. This paper is easy to follow and overall well-structured.

2. I think the problem this paper tries to tackle is interesting for the community. Designing an efficient attention mechanism in Transformers for vision tasks is definitely an important problem. Especially, computing attention within token pyramids following a quardtree structure is novel to me, which has not been studied in previous work.

3. The experiments are comprehensive and strong. It is good to see the comparisons on four different computer vision tasks which cover both self-attention and cross-attention. Although the improvement in accuracy seems minor to me, the reduction in computation is notable.

Cons.

Overall, I have some concerns about the experiment, and the following points might be worthy to study in the rebuttal period.

1. Spare attention usually leads to a faster convergence rate but higher final losses compared to full attention. I wonder if this still holds in QuadTree Attention. It is interesting to see the training curve of the proposed method compared with ViT or Swin.

2. I wonder if the number of tokens in the finest resolution will affect the model performance. E.g., what is the current patch size of the input image, and will it lead to performance improvement if a larger or smaller patch size is chosen?

3. It is also worthy to explore if variants of larger (smaller) size in model parameters will lead to performance improvement, and how FLOPs change in these variants.

**Summary Of The Paper:**

This paper aims to address the quadratic computational complexity of vanilla Transformers. The key idea is to build token pyramids and computes attention in a coarse-to-fine manner, which reduces the computational complexity to linear. The resulting attention paradigm forms a quadtree structure, and two variants, QuadTree-A and QuadTree-B, are further proposed to improve message aggregation. Experiments are conducted on four different computer vision tasks involving either self-attention or cross attention. Results show that the proposed approach matches the state of the art with much fewer FLOPs and model parameters.

**Summary Of The Review:**

The paper studied a challenging and important problem in efficient attention mechanisms. I think the overall method is novel and the experimental results are promising, which leads me to a positive rating. I hope the authors can further address my concerns in the rebuttal.

---

> ### Author Response · Authors · 2021-11-16
> **Reponse to R3**
>
> We appreciate for the constructive suggestions.
>
> 1. Convergence comparison with PVTv2 and Swin.
>
>     a) For the cross attention tasks, we provide training loss and testing accuracy curve in Fig.3 in the supplementary. QuadTree-B has larger training loss at first than PVT, then smaller loss in the final. But our method almost always has higher accuracy.
>
>     b)For the self attention tasks, we provide training and validation accuracy curve of PVTv2-B0, Swin and our method in Figure 4. Our method achieves lower training loss and better validation accuracy all the time.
>
> 2. *patch size*.
>
>     We down-sample the image to 1/4 by a $7\times7$ convolution with a stride of 4, which means $4 \times 4$ sized patches are used. This is quite a common setting, ResNet/Swin/PVT all use this setting.
>
>  3.*model size.*
>
>     As shown in Tab. 3 and Tab. 4, we provide b0, b1, b2 networks with different model sizes, with a parameter number of 3.4M, 13.6M and 24.2M, a flop count of 0.6G, 2.2G, and 4.3G on ImageNet, and the top 1 accuracy is 71.9, 79.7 and 82.6. Due to limited resource, we will release results for larger models in the future.

---

> > ### Author Response · Authors · 2021-11-22
> > **The influence of patch size**
> >
> > The influence of patch size: we conduct an experiment using a bigger patch size. Specifically, we down-sample the image to 1/8 by a  convolution with a stride of 8 and other architectures are kept the same as QuadTree-B-b0. We observe a performance drop from 71.9 to 67.2.  Using a smaller patch size (e.g., down-sample to 1/2) will result in a significant computation increase and currently, we do not have enough resources for this experiment.

---

> > > ### Comment · Reviewer_PDon · 2021-11-30
> > > **Thank you for your response**
> > >
> > > Thank the authors for their response. After reading other reviews and the corresponding responses, I believe my concerns and most primary concerns of other reviewers have been addressed. Thus I would like to keep my initial rating and recommend acceptance for this paper.

---

### Official Review · Reviewer_sHEA · 2021-11-03

**Correctness:** 3
**Technical Novelty And Significance:** 2
**Empirical Novelty And Significance:** Not applicable
**Recommendation:** 5
**Confidence:** 4

**Main Review:**

## Strengths

- The idea of the paper is clear and easy to follow up.
- The effectiveness of the proposed approach is verified on different tasks.

## Weaknesses
- Some important related works are missing. The previous work [1] also proposed a multi-level attention approach to efficiently balance the short-distance and long-distance attention, which proposed the same way to generate the multi-level or fine and coarse tokens. From this point, the idea of quadtree-b attention is very similar to [1]. However, the authors did not mention and discuss the relation to this very related work. I suggest the authors carefully discuss the relation [1].

- The paper proposed an efficient way to process long-distance and short-distance attention. There should be some baselines we should compare with, 1) the vanilla attention mechanisms, 2) some other efficient attention mechanisms, e.g. shifted windows attention in Swin-transformer, focal attention in Focal Transformer[1], sampled K,V in PVT. However, the authors only compared with the PVT. I'd like to see more solid apple-to-apple comparisons and discussions on that.

## Questions:
- How about the real runtime when applying quadtree attention to PVT architecture?
- As I posted above, I think the quadtree attention mechanism is a general attention mechanism. I'd like to know how about the performance when applying to other transformer-based architecture, e.g. Swin-transformer, ViT, et. al.
- The authors argue that Swin-transformer restricts the attention in the local windows. However, by shifting the windows in the Swin Transform, it enables that information exchange between windows for Swin. From this point, I don't think that Swin-transformer limits the attention whin local windows.


[1] Focal Self-attention for Local-Global Interactions in Vision Transformers.

**Summary Of The Paper:**

The paper proposed an attention approach to handle the global or long-range attention by leveraging the idea of quadtree structure, meanwhile reducing the quadratic complexity of original attention operation to linear. Experiments are performed on several tasks, e.g. feature matching, image classification, and objection detection. Superior results are achieved on these tasks.

**Summary Of The Review:**

The paper proposed a clear method to handle long-distance and short-distance attention. However, important related work is missing in the discussion. As mentioned above, I think more discussions and experiments should be added to support the claim. I am leaning to reject the paper if the authors can't address my concerns.

---

> ### Author Response · Authors · 2021-11-16
> **Responses to R2**
>
> We thank for the valuable comments.
> 1. Comparison with Focal attention:
>
>      Focal attention is a concurrent work that also aims to capture both global and local attention in one single module, but is quite different from our work:
>
>      a) Focal attention only down-samples the key and value tokens and does not down-sample the query tokens when building multiscale attention, similar to the Poolingformer[1]. In contrast, we down-sample all these tokens. This can bring more efficiency in memory and computation since coarse level messages can be shared for the query sub-patches in the same patch, as shown in Fig. 1.
>
>      b) Attention score and message generation. Focal attention concatenates all tokens from different levels in a single Softmax operator to compute the attention score and message. While our method computes the attention score and message separately at different levels. Thus, query tokens can share the same coarse level messages to reduce overhead.
>
>      c) Focal attention actually uses two-level attention in the released code. For high-resolution tasks like object detection, it covers about $1/6\approx(15\times7\times4)^2 / (800\times1333) $ size of the image in the first stage, even if windows size has been increased adaptively for COCO. While our method always covers the whole image. Increasing the windows size or window level will result in larger memory consumption, although it already requires 2x memory than ours as shown in Tab. 5.
>
>      d) Our work aims to design a general vision transformer not only for self attention tasks but also cross attention tasks. For cross attention tasks, the most related pixels in another image can be far away from the pixel itself, e.g., in the wide-baseline feature matching tasks shown in our paper. Thus, restricting the attention computation in a nearby window like Swin and Focal attention in a single attention computation might not be optimal for these tasks. Instead, we use the top K selection mechanism, which allows the query tokens to attend dynamic regions in fine levels.
>
>      e) We give a comparison with Focal attention in classification and detection under the PVTv2-b0 setting. As shown in Tab.5, we achieve higher results especially in detection, where high-resolution images are required, with an improvement of 1.0 AP.
>
>      f) Focal attention is an arxiv paper when we submitted our paper into ICLR, and we did not notice it. We add discussions about it in the latest paper in Section 2, 'Vision Transformer'.
>
> 2. Comparison with other transformers.
>
>     (1) The vanilla attention mechanisms. We present a comparison with standard transformers in the stereo matching tasks. It achieves similar performance but is much more efficient. Due to limited resources, we did not give a comparison with the vanilla transformer on self attention task, e.g. ViT, since we have already compared with other recent efficient transformers. Besides, ViT is not suitable for high-resolution tasks like object detection.
>
>     (2) Comparison with other efficient transformers.
>
>     a) Linear approximate attention. We give a comparison with Linear transformer on feature matching tasks as shown in Tab. 1. Our method is 2.7 higher in AUC@10$^{\circ}$.
>
>     b) Swin Transformer. As shown in Tab. 5 of the latest submission, our method can achieve better results than Swin on both ImageNet and COCO, with an improvement of 1.4\% and 3.2\% AP respectively, which shows the effectiveness of global attention.
>
> 3. Running time.
>
>     Currently, we only implement a naive CUDA codes without many optimizations, and it is not as efficient as the well-optimized dense GPU matrix operation. We test the running time of Retinanet under PVTv2-b0 architecture. We test the speed under a batch size of 10. For PVTv2-b0, the running time is 0.022s to forward one image and for Quadtree-b0, the running time is 0.038s for forwarding one image on average. However, Quadtree-B-b0 has much lower memory usage than PVTv2-b0. Quadtree-B-b0 consumes about 339 MB memory, while PVTv2-b0 consumes about 574 MB memory for one $800\times 1333$ image.
> Further speed-up can be achieved by more engineering optimizations, such as {https://openreview.net/forum?id=agBJ7SYcUVb}.
>
> 4. Comparison under other transformer  architecture.
>
>     As shown in Tab.8 in Appendix. E, we run Quadtree-B attention using Swin transformer architecture with the released codes. Our Quadtree-B attention obtains 82.2\% top 1 accuracy, 0.9\% higher than Swin transformer on ImageNet.
>
> 5. *Swin-transformer limits the attention within local windows*
>
>    By saying this, we mean Swin computes attention in a local window in a single operation. Swin has to stack more blocks to capture long-range connections like CNN. We make it clear in the latest paper.
>
> [1] Poolingformer: Long Document Modeling with Pooling Attention. ICML 2021.

---

> > ### Author Response · Authors · 2021-11-22
> > **Comparison with other transformers**
> >
> > 1. We have updated comparison results with other transformers in the PVTv2-b0 setting (Table 5) and Swin transformer setting (Table 8). For more fair comparisons, we equip all transformers with the same position encoding methods (Table 5). In both settings, we achieve higher results than PVTv2, Swin and Focal Transformer.
> >
> > 2. Although we have already given comparisons with Focal Transformer thoroughly in Table 5 and Table 8, it should also be noted that, according to the policy of ICLR review, it is not necessary to compare work available in Arxiv or publish within 4 months.
> >
> > |                 | ImageNet-1K |           | COCO (RetinaNet) |      |      |      |
> > |--------|---------------|---------------|------------------|------|------|------|
> > |                      | Flops (G)   | Top-1(%) | Mem.  (MB)       | AP   | AP50 | AP75 |
> > | PVTv2                | 0.6         | 70.5      | 574              | 37.2 | 57.2 | 39.5 |
> > | PVTv2+LePE           | 0.6         | 70.9      | 574              | 37.6 | 57.8 | 39.9 |
> > | Swin                 | 0.6         | 70.5      | 308              | 35.3 | 54.2 | 37.4 |
> > | Swin+LePE            | 0.6         | 70.7      | 308              | 35.8 | 55.3 | 37.7 |
> > | Focal Attention      | 0.7         | 71.6      | 732              | 37.5 | 57.6 | 39.5 |
> > | Focal Attention+LePE | 0.7         | 71.5      | 732              | 37.1 | 57.0 | 39.4 |
> > | QuadTree-B           | 0.6         | 71.9      | 339              | 38.5 | 58.8 | 41.1 |
> >
> > Table 5: Comparison with PVTv2, Swin and Focal transformer under PVTv2-b0 settings.
> >
> > |            | ImageNet   |           |          | COCO (RetinaNet) |      |      |
> > |------------|------------|-----------|----------|------------------|------|------|
> > |            | Param. (M) | Flops (G) | Top1 (%) | AP               | AP50 | AP75 |
> > | Swin-T     | 29         | 4.5       | 81.3     | 42.0             | \    | \    |
> > | Focal-T    | 29         | 4.9       | 82.2     | 43.7             | \    | \    |
> > | Quadtree-B | 30         | 4.6       | 82.2     | 44.6             | 65.8 | 47.7 |
> >
> > Table 8: Comparison under Swin-T settings.

---

> ### Comment · Reviewer_sHEA · 2021-11-30
> **Final decision**
>
> The author solved some of my concerns. I'd like to upgrade to weak acceptance. I encourage the authors to have more discussion on the relation with Focal Transformers in the main paper since it's a very related concurrent work.

---

### Official Review · Reviewer_jF1Z · 2021-11-03

**Correctness:** 3
**Technical Novelty And Significance:** 3
**Empirical Novelty And Significance:** 3
**Recommendation:** 6
**Confidence:** 4

**Main Review:**

Pros:
1. The multi-scale design in quadtree attention is reasonable:
The quadtree attention can model long-range interactions using coarse tokens from less relevant regions, while establishing dense interactions using fine tokens from informative local regions. Thus, the multi-scale structure (in a coarse-to-fine manner) in quadtree attention provides and efficient way to capture both long-range and local interactions. A further modified version of quadtree attention (QuadTree-B) introduces additional weights and overlapping regions, reducing the effect from inaccurate attention score on coarse tokens.

2. The quadtree attention can be used in both self-attention and cross-attention, which leads to a wider applicability.

3. Empirical performance demonstrates the effectiveness of the proposed approach: On feature matching tasks, the proposed quadtree achieves significant improvement on AUC of camera pose errors. On stereo matching and object detection, it is observed that the computation (Flops) is reduced by a large margin with on-par performance.

4. The paper is well written and easy to follow.

Cons / Questions / Suggestions:
1. From the technical contribution aspect, the quadtree structure is not brand new. It is arguably true that the proposed approach is one of the earliest attempts to introduce quadtree structure into the attention mechanism. However, since some recent work [1] also employs coarse-to-fine structure in attention, it would be better to have some comparison with such work.
[1] Focal Self-attention for Local-Global Interactions in Vision Transformers, NeurIPS 2021

2. The design of pyramid tokens is unclear:
  - (1) In Figure 1, Top-2 patches are used to compute finer attention. I wonder why in level 2 there are two sub-patches (green and yellow)? From the position of red patch in level, we should only have the green patch in level 2 (which could be consistent with Figure 2). In addition, why the two green patches in level 2 of image B have the same location in the 2x2 grid (both are lower left)? The same pattern is also observed in red patches in level 3.
  - (2) It remains unclear that how the pyramid tokens are generated: In Page 4-5, it mentions "we construct L-level pyramids for query and value V tokens respectively by 2x2 average pooling". Does it mean the coarse tokens are always average pooled fine tokens at one level lower? Is there any additional patch embedding layer introduced?
  - (3) In level 1, top K patches are selected, which leads to 4K sub-patches in level 2. Does it uniformly sample top K sub-patches from 4K sub-patches, or each 2x2 sub-patches from coarse patch must have one sampled sub-patch?

3. In the empirical experiments, the quadtree attention does not bring much efficiency improvement on the ImageNet classification: It has very similar #parameters and #flops compared with its baseline (PVTv2), which is quite different from the object detection task. I wonder what is the reason behind such as big gap.

4. Typos
- Equation (1): Should be V instead of V^T
- Figure 2: It would be better to use \mathbf{m}_i^1 instead of m_i^1 for a consistent notation in the text.
- Line below Eq (4): Should s_ij^1 be s_ij^0?

**Summary Of The Paper:**

This paper proposes a new attention mechanism in vision transformers, which is called quadtree attention. This quadtree attention mechanism builds token pyramids and computes the attention in a coarse-to-fine manner. At each level, only top K regions with the highest attention scores from query and key are selected for further attention computation on finer tokens, while other regions simply use (current) coarse attention as a part of output. In the empirical study, the proposed quadtree attention achieves good performance on a wide range of vision tasks (e.g. feature matching, stereo matching, classification and object detection) with less computation.

**Summary Of The Review:**

In summary, I think the proposed quadtree attention is a reasonable efficient design of attention mechanism. It builds coarse and fine tokens, which is able to model both long-range and local interactions. Empirical results also demonstrate that the proposed approach achieves good performance with less computation on many vision tasks. However, I still hold some concerns in technical contribution, unclear design of the pyramid tokens, and performance on classification task. Thus, I would like to rate this paper as "marginally above the acceptance threshold".

---

> ### Author Response · Authors · 2021-11-16
> **Responses to R1**
>
> We thank for the very detailed and constructive suggestions.
>
> 1. Comparison with Focal attention:
> Focal attention is a concurrent work that also aims to capture both global and local attention in one single module, but is quite different from our work:
>
>      a) Focal attention only down-samples the key and value tokens and does not down-sample the query tokens when building multiscale attention, similar to the Poolingformer[1]. In contrast, we down-sample all these tokens. This can bring more efficiency in memory and computation since coarse level messages can be shared for the query sub-patches in the same patch, as shown in Fig. 1.
>
>      b) Attention score and message generation. Focal attention concatenates all tokens from different levels in a single Softmax operator to compute the attention score and message. While our method computes the attention score and message separately at different levels. Thus, query tokens can share the same coarse level messages to reduce overhead.
>
>      c) Focal attention actually uses two-level attention in the released code. For high-resolution tasks like object detection, it covers about $1/6\approx(15\times7\times4)^2 / (800\times1333) $ size of the image in the first stage, even if windows size has been increased adaptively for COCO. While our method always covers the whole image. Increasing the windows size or window level will result in larger memory consumption, although it already requires 2x memory than ours as shown in Tab. 5.
>
>      d) Our work aims to design a general vision transformer not only for self attention tasks but also cross attention tasks. For cross attention tasks, the most related pixels in another image can be far away from the pixel itself, e.g., in the wide-baseline feature matching tasks shown in our paper. Thus, restricting the attention computation in a nearby window like Swin and Focal attention in a single attention computation might not be optimal for these tasks. Instead, we use the top K selection mechanism, which allows the query tokens to attend dynamic regions in fine levels.
>
>      e) We give a comparison with Focal attention in classification and detection under the PVTv2-b0 setting. As shown in Tab.5, we achieve higher results especially in detection, where high-resolution images are required, with an improvement of 1.0 AP.
>
>      f) Focal attention is an arxiv paper when we submitted our paper to ICLR, and we did not notice it. We add discussions about it in the latest paper in Section 2, 'Vision Transformer'.
>
>
> 2. The design of token pyramid.
>
>    (1) Questions about Figure 1
>
>    a) *there are two sub-patches (green and yellow) in the level 2.* We show two sub-patches in level 2 to indicate that these two sub-patches share the same message computation in level 1. They have the same kNN patches, attention score and message. For level 3, we did not show more sub-sub-patches because of the complexity.
>
>    b) *the two green patches in level 2 of image B have the same location.* This is a coincidence, they can be at random positions. We update Figure 1 to get rid of confusion.
>
>    c) *Does it uniformly sample top K sub-patches from 4K sub-patches*. Yes, we sample top K sub-patches according to the attention score, no other restrictions are applied.
>
>    (2) Pyramid tokens generation.
>
>     For QuadTree-A, we use average pooling for query, key and value tokens. For QuadTree-B, we use average pooling for query and key tokens, but for the value pyramid, we use convolution with stride 2 and normalization activation. This is also used in PVT and Focal transformer to down-sample key and value tokens. The total number of parameters is still less than other baselines, such as PVTv2. This value branch is useful for self attention tasks, but not for cross attention tasks. So only average pooling is used for cross attention tasks.
> We make these issues clear in the latest submission.
>
>
> 3. Efficiency improvement on object detection.
>
>     The image resolution in COCO detection is much higher than ImageNet classification, $800 \times 1333$ vs $224 \times 224$. PVTv2 down-samples tokens by a fixed stride, which means the query token needs to compute attention with more key tokens when resolution is higher. While our method always computes attention with fixed top K tokens. Therefore, our method is more efficient when using higher resolution images.
>
> 4. Typos.
>
>     a) *Equation (1): Should be V instead of V^T*: Yes.
>
>     b) *Figure 2: It would be better to use \mathbf{m}_i^1 instead of m_i^1 for a consistent notation in the text.*: Yes.
>
>     c) *Line below Eq (4): Should s_ij^1 be s_ij^0?*: No, we start from 1 as shown in Eq. 2.
>
>     We fix these things in the latest submission.
>
> [1] Poolingformer: Long Document Modeling with Pooling Attention. ICML 2021.

---

> > ### Author Response · Authors · 2021-11-22
> > **Comparison with other transformers**
> >
> > 1. We have updated comparison results with other transformers in the PVTv2-b0 setting (Table 5) and Swin transformer setting (Table 8). For more fair comparisons, we equip all transformers with the same position encoding methods (Table 5). In both settings, we achieve higher results than PVTv2, Swin and Focal Transformer.
> >
> > 2. Although we have already given comparisons with Focal Transformer thoroughly in Table 5 and Table 8, it should also be noted that, according to the policy of ICLR review, it is not necessary to compare work available in Arxiv or publish within 4 months.
> >
> > |                 | ImageNet-1K |           | COCO (RetinaNet) |      |      |      |
> > |--------|---------------|---------------|------------------|------|------|------|
> > |                      | Flops (G)   | Top-1(%) | Mem.  (MB)       | AP   | AP50 | AP75 |
> > | PVTv2                | 0.6         | 70.5      | 574              | 37.2 | 57.2 | 39.5 |
> > | PVTv2+LePE           | 0.6         | 70.9      | 574              | 37.6 | 57.8 | 39.9 |
> > | Swin                 | 0.6         | 70.5      | 308              | 35.3 | 54.2 | 37.4 |
> > | Swin+LePE            | 0.6         | 70.7      | 308              | 35.8 | 55.3 | 37.7 |
> > | Focal Attention      | 0.7         | 71.6      | 732              | 37.5 | 57.6 | 39.5 |
> > | Focal Attention+LePE | 0.7         | 71.5      | 732              | 37.1 | 57.0 | 39.4 |
> > | QuadTree-B           | 0.6         | 71.9      | 339              | 38.5 | 58.8 | 41.1 |
> >
> > Table 5: Comparison with PVTv2, Swin and Focal transformer under PVTv2-b0 settings.
> >
> > |            | ImageNet   |           |          | COCO (RetinaNet) |      |      |
> > |------------|------------|-----------|----------|------------------|------|------|
> > |            | Param. (M) | Flops (G) | Top1 (%) | AP               | AP50 | AP75 |
> > | Swin-T     | 29         | 4.5       | 81.3     | 42.0             | \    | \    |
> > | Focal-T    | 29         | 4.9       | 82.2     | 43.7             | \    | \    |
> > | Quadtree-B | 30         | 4.6       | 82.2     | 44.6             | 65.8 | 47.7 |
> >
> > Table 8: Comparison under Swin-T settings.

---

> > > ### Comment · Reviewer_jF1Z · 2021-11-28
> > > **Thank you for your detailed response!**
> > >
> > > Thank you for your detailed response! The response has answered most of my questions. However, I still have a bit question about the feedback for Typo c):
> > >
> > > *c) Line below Eq (4): Should s_ij^1 be s_ij^0?: No, we start from 1 as shown in Eq. 2.*
> > >
> > > I do know that the level starts from 1. However, does it mean that s_ij^1 is always 1 (i.e. the attention score between query and key tokens at level 1 is always 1)? Then in eq (3), m^1_i is simply the average of all value vectors? Hope the authors can provide some additional clarification here. Thanks!

---

> > > > ### Author Response · Authors · 2021-11-28
> > > > **Thank you for your correction**
> > > >
> > > > Thank you for your correction. We made a mistake in the main paper. $s_{ij}^1$ should be $s_{ij}^0$ and $s_{ij}^0=1$. Level 0 does not exist, and we represent the score as $s_{ij}^0=1$ for the completeness of equation 4.

---

> > > > > ### Comment · Reviewer_jF1Z · 2021-11-30
> > > > > **Thank you for your clarification**
> > > > >
> > > > > Thank you for your clarification. I would like to keep my weak accept rating.

---

### Author Response · Authors · 2021-11-16
**Paper Update**

We thank the reviewers for the valuable comments. We first present the guideline about our paper update, and then answer the questions of each reviewer.

1. Following PVTv2, we did not use positional encoding operation in the attention computation block for self attention tasks, while other works like Swin and Focal transformer already have positional encoding module.
After the paper submission, we augment our network with a multiscale positional encoding module (MPE), which is adapted from the locally-enhanced positional encoding [1]. The details are illustrated in Section 3.2, 'Multiscale position encoding'.
For objection detection, the AP can be improved by $1.3 - 1.6$ with our MPE module.
Results and more ablation studies are updated in the paper.

2. More fair comparisons with other transformers. Due to limited time and resources, we give a comparison with PVTv2, Swin, Focal attention using a small model. More specifically, we replace the attention modules of PVTv2-b0 with these attention modules. Results are shown in Table. 5 of the latest paper.


[1] Cswin transformer: A general vision transformer backbone with cross-shaped windows. arXiv 2021.

---

### Author Response · Authors · 2021-11-22
**Paper Update v2**

We make the following changes for the main paper:
1. We update object detection results in Swin transformer setting in Table 8.
In object detection, Quadtree attention is 2.6 higher than Swin. When compared with Focal Transformer, Quadtree attention obtains the same top 1 accuracy in image classification with fewer flops and 0.9 higher AP in object detection.

2. We update comparison results with other transformers in the PVTv2-b0 setting.
For fair comparisons, we adopt the same positional encoding method LePE[1] for PVTv2, Swin and Focal transformer in Table 5. It is shown that quadtree attention obtain the best performance no matter what position encoding method is adopted.

[1] Cswin transformer: A general vision transformer backbone with cross-shaped windows. arXiv 2021.

---

### Decision · Program_Chairs · 2022-01-20

**Decision:**

Accept (Poster)

**Comment:**

The paper proposes an efficient attention variant inspired by quadtrees, for use in vision transformers. When applied to several vision tasks, the approach leads to better results and/or less compute.

The reviews are all positive about the paper, after taking into account the authors' feedback (one reviewer forgot to update their official rating, apparently). They point out that the idea is reasonable and the empirical evaluation is thorough and convincing, with good gains on several tasks and datasets.

Overall, I recommend acceptance.